# The IBA-ISMO Method for Rolling Bearing Fault Diagnosis Based on VMD-Sample Entropy

**DOI:** 10.3390/s23020991

**Published:** 2023-01-15

**Authors:** Deyu Zhuang, Hongrui Liu, Hao Zheng, Liang Xu, Zhengyang Gu, Gang Cheng, Jinbo Qiu

**Affiliations:** 1China Coal Technology and Engineering Group Shanghai Company Ltd., Shanghai 200030, China; 2School of Mechatronic Engineering, China University of Mining and Technology, Xuzhou 221116, China

**Keywords:** variational mode decomposition, sample entropy, sequence minimum optimization algorithm, fault diagnosis

## Abstract

Rolling bearings are important supporting components of large-scale electromechanical equipment. Once a fault occurs, it will cause economic losses, and serious accidents will affect personal safety. Therefore, research on rolling bearing fault diagnosis technology has important engineering practical significance. Feature extraction with high price density and fault identification are two keys to overcome in the field of fault diagnosis of rolling bearings. This study proposes a feature extraction method based on variational modal decomposition (VMD) and sample entropy and also designs an improved sequence minimization algorithm with optimal parameters to identify the fault. Firstly, a variational modal decomposition system based on vibration signals is designed, and the sample entropy of the components is extracted as the eigenvalue of the signal. Secondly, in order to improve the accuracy of fault diagnosis, the sequence minimum optimization algorithm optimized by the bat algorithm is used as the classifier. Certainly, the traditional bat algorithm (BA) and the sequence minimum optimization algorithm (SMO) are improved, respectively. Therefore, a fault diagnosis algorithm based on IBA-ISMO is obtained. Finally, the experimental verification is designed to prove that the algorithm model has a good state recognition rate for bearings.

## 1. Introduction

The common faults of electromechanical equipment can be divided into electrical faults and mechanical faults. Electrical faults are often caused by circuit aging, component damage, etc., while mechanical faults refer to the phenomenon that electromechanical equipment loses or reduces its specified functions and cannot continue to operate due to some inevitable damage. Mechanical faults have a serious impact on the safety state of electromechanical equipment [1,2,3]. On the one hand, the root cause of faults is complex and the evolution time is long; on the other hand, once the mechanical faults lead to an accident, the impact and consequences are unpredictable.

Many mechanical faults are reflected in the form of vibration, and the vibration signals contain rich information, which can quickly and directly reflect the operating status of critical parts in major equipment such as bearings [4,5,6]. It is very necessary to carry out vibration monitoring and fault diagnosis. However, extracting feature information with high valence density and designing a classification space that is suitable for strong nonlinear and non-stationary information are urgent problems to be solved in the field of fault diagnosis, and even in the field of machine learning. Reference [7] proposed an adaptive boundary determination method based on empirical wavelet transform and applied it to fault detection of high-speed train wheelset bearings. Park et al. [8] proposed a minimum variance cepstrum based on cepstral analysis, which avoided the influence of the system frequency and the selection of the resonance band, and realized the detection of early faults of the rotating parts. Borghesani proposed a method of whitening the signal using cepstrum [9]. [10] proposed an improved empirical mode decomposition method to effectively extract the fault features of rolling bearings. In view of the nonlinear and non-stationary characteristics of the vibration signal of planetary gears, [11] proposed a feature extraction method of initial fault based on ensemble empirical mode decomposition (EEMD) and adaptive stochastic resonance (ASR), which provided the strong initial fault diagnosis of planetary gears against noise background. However, in practical application, the wavelet decomposition method has problems such as difficulty in selecting the wavelet basis function [12], and the empirical mode method often has the problem of end-point effects and mode mixing, which presents some challenges regarding the extraction of fault features. As a time-frequency domain analysis method, variational mode decomposition (VMD) has better adaptability than other analysis methods. This method combines the classic Wiener filtering, Hilbert transform and frequency mixing in mathematical theory. Based on these advantages, the number of self-determined modal components and the lower time complexity are realized, and the non-stationary original signal can be decomposed into relatively stationary subsequences containing multiple frequency domains by VMD. Wang et al. [13] proposed the characteristic parameter of spectral kurtosis entropy (SKE) and combined it with VMD to realize the feature extraction of the bogie vibration signal under variable working conditions. [14] proposed a sparse VMD (sparsity-oriented VMD) method, which effectively extracted encoder information and realized gear fault diagnosis.

Fault identification is also an important step for establishing the correlation between fault features and class labels. Fan et al. [15] proposed a high-performance SVM multi-feature fusion and self-tuning particle swarm optimization algorithm. The method extracted multi-dimensional fault features by EMD. Then, the multi-dimensional parameters of the high-performance SVM were configured by adjusting the particle swarm optimization algorithm, which has improved the effectiveness in bearing fault detection and classification. David E. Runelhart et al. [16] proposed the back propagation (BP) neural network algorithm, which constituted a multi-layer feedforward perceptron to solve the problem of connection weight learning in the hidden layer of the multi-layer neural network. Lu et al. [17] proposed an improved feature selection and neural network classification algorithm for the problem of rotating machinery fault diagnosis. The study extracted the time domain and frequency domain features of the whole machine under multiple working conditions and used an optimized backpropagation neural network algorithm for fault diagnosis. He et al. [18] proposed a bearing fault diagnosis method based on a Gaussian constrained Boltzmann machine, which takes the envelope spectrum of the resampled data directly as a feature vector to represent the bearing fault. Wang et al. [19] proposed an intelligent bearing fault diagnosis method that combined the symmetric point pattern representation and the compressed excitation convolutional neural network model for the problems of fault visualization and automatic feature extraction.

Based on the above analysis, this paper intends to use variational modal decomposition (VMD) and sample entropy for signal decomposition and feature extraction of vibration signals. The improved sequence minimum optimization algorithm has been chosen as the pattern recognition method in this study.

The rest of the article is organized as follows: Section 2 focuses on the feature extraction method based on VMD-Sample Entropy and the sequence minimum optimization algorithm after optimizing parameters. Section 3 verifies the validity of the fault diagnosis model proposed in this study through experiments and conducts necessary analysis on the experiments. The fourth part summarizes the research of the full study.

## 2. The Enhanced Fault Diagnosis Method

The study plans to decompose the original signal by VMD to obtain the intrinsic mode component (IMF) and to extract the sample entropy of each component as the eigenvalue of the bearing fault. The training set is input into the improved sequence minimum optimization algorithm (ISMO) model for training. At the same time, the penalty factor and Gaussian kernel parameters of the ISMO are optimized by an improved bat algorithm (IBA). The fault diagnosis model established in this study is shown in Figure 1.

### 2.1. Feature Extraction Method Based on VMD-Sample Entropy

Signal denoise and feature extraction often play a vital role, respectively. In this study, the variable scale processing method, known as variational mode decomposition (VMD), is proposed. VMD satisfies the self-adaptability of the decomposition model in a non-recursive way and transforms the complex signal decomposition problem in the time domain and frequency domain into a mathematical model for the solution so as to avoid the end effect and restrain the mode mixing caused by noise, and an ideal optimal result of signal decomposition is naturally obtained.

After the optimal result of signal decomposition is obtained through the variational model, in order to better carry out fault diagnosis, it is necessary to extract the most prominent feature information from the signal decomposition results. In this study, the sample entropy is selected as the feature of the vibration signal. As a new algorithm based on approximate entropy algorithm, the physical meaning of sample entropy can be expressed as calculating the probability of the change of time series caused by the change of data bits.

#### 2.1.1. Variational Mode Decomposition

Constructing a constrained variational model.

Firstly, the analytic signal $u_{k}$ of the original signal is obtained by Hilbert transform of the real mode function $u_{k}^{+}$.
(1)$$u_{k}^{+}(t)=\left(\delta \left(t\right)+\frac{j}{\pi t}\right)*u_{k}(t)$$

In Formula (1), *t* and δ(t) denote time and influence function, respectively.

The analytical signal $u_{k}^{+}$ is mixed with each estimated center frequency, and the spectrum of each mode is modulated to the corresponding fundamental frequency band as follows:(2)$$u_{k}^{m}(t)=\left(\delta \left(t\right)+\frac{j}{\pi t}\right)*u_{k}(t)e^{-jw_{k}t}$$

Finally, by calculating the $L_{2}$ norm of the time gradient, the effective value of the modal component bandwidth can be calculated as follows:(3)$$\Delta w={\left\|\partial_{t}\left[\left(\delta \left(t\right)+\frac{j}{\pi t}\right)*u_{k}(t)e^{-jw_{k}t}\right]\right\|}_{2}^{2}$$

Therefore, the bandwidth of the modal components of each frequency can be expressed as Formula (4):(4)$$\begin{array}{c} \underset{\left\{u_{k}\right\},\left\{w_{k}\right\}}{\min}\left\{\sum_{K=1}^{K}{\left\|\partial_{t}\left[\left(\delta \left(t\right)+\frac{j}{\pi t}\right)*u_{k}(t)e^{-jw_{k}t}\right]\right\|}_{2}^{2}\right\}\\ \sum_{K=1}^{K}u_{k}\left(t\right)=f\left(t\right) \end{array}$$

In Formula (4), $\left\{u_{k}\right\}=\left\{u_{1},\ldots ,u_{k}\right\}$ represents the IMF components obtained by VMD; $\left\{w_{k}\right\}=\left\{w_{1},\ldots ,w_{k}\right\}$ represents the central frequency of IMF, f(t) is the original input signal.

(2)Solving constrained variational model

Formula (4) is constructed as a Lagrangian expression by adding the quadratic penalty factor and the Lagrange operator λ(t).
(5)$$\begin{array}{c} L_{\left(\{u_{k}\},\{w_{k}\},\lambda \right)}:=\alpha \{\sum_{k}\|\partial_{t}[(\delta (t)+\frac{j}{\pi t})\times u_{k}(t)]{e^{-jw_{k}t}\|}_{2}^{2}\}+\|f\left(t\right)-{\sum_{k}u_{k}(t)\|}_{2}^{2}\\ +\langle \lambda (t),f\left(t\right)-\sum_{k}u_{k}(t)\rangle \end{array}$$

In Formula (5), the appropriate penalty factor is selected to ensure that the reconstruction accuracy is high enough under variable working conditions, and the Lagrangian operator λ(t) is introduced to make the solution of Formula (5) theoretical and rigorous.

The alternating direction multiplier algorithm is introduced to solve the above variational problems. The main idea is to obtain the saddle point of the extended Lagrangian expression by alternately updating the parameters $u_{k}\left(t\right)$, $w_{k}\left(t\right)$ and $\lambda_{k}\left(t\right)$. The updated Formula is as follows (6):(6)$$\begin{array}{c} u_{k}^{n+1}\left(t\right)=\arg\underset{u_{k}\in X}{\min}\{\alpha {\sum_{K=1}^{K}\left\|\partial_{t}\left[\left(\delta \left(t\right)+\frac{j}{\pi t}\right)*u_{k}\left(t\right)e^{-jw_{k}t}\right]\right\|}_{2}^{2}\\ +{\left\|f\left(t\right)-\sum_{i}u_{i}\left(t\right)+\frac{\lambda \left(t\right)}{2}\right\|}_{2}^{2}\} \end{array}$$

Under the condition of $L_{2}$ norm, Equation (6) is transformed into the frequency domain by Fourier isometric transform, and the variable w in the equation is replaced by the updated $w-w_{k}$. According to the Hermitain symmetry theorem, the expression of the *k* eigenmode function (IMF) is obtained as follows:(7)u∧kn+1(w)=f∧(w)−∑u∧i<kin+1(w)−∑u∧i>kin(w)+λ∧(w)21+2α(w−wn)2

The central frequency expression of the updated modal IMF is:(8)$${\overset{\wedge }{w}}_{k}^{n+1}=\frac{\int_{0}^{\infty }w{\left|{\overset{\wedge }{u}}_{k}^{n+1}\left(w\right)\right|}^{2}dw}{\int_{0}^{\infty }{\left|{\overset{\wedge }{u}}_{k}^{n+1}\left(w\right)\right|}^{2}dw}$$

The updated expression of all non-negative center frequencies is w≥0. and the updated expression of operator $\lambda^{n+1}$ is:(9)$${\overset{\wedge }{\lambda }}^{n+1}\left(w\right)={\overset{\wedge }{\lambda }}^{n}\left(w\right)+\tau \left[\overset{\wedge }{f}\left(w\right)-\sum_{k}{\overset{\wedge }{u}}_{k}^{n+1}\left(w\right)\right]$$

Summing up the above description, the decomposition process of VMD algorithm can be summarized as follows:

(1)Initialize the value of $u_{k}^{1}\left(t\right)$, $w_{k}^{1}\left(t\right)$ and $\lambda_{k}^{1}\left(t\right)$, *n* is 0.(2)Set the out-of-loop condition: n=n+1.(3)Update $u_{k}\left(t\right)$ and $w_{k}\left(t\right)$ until the number of intrinsic mode decomposition of the original sample meets the preset number of the decomposition, ending the current internal cycle.(4)Get a new $\lambda_{k}\left(t\right)$ license.(5)Give the jump condition ε as the operator precision, and the $\frac{\sum_{k}{\left\|u_{k}^{n+1}-u_{k}^{n}\right\|}_{2}^{2}}{{\left\|u_{k}^{n}\right\|}_{2}^{2}}<\varepsilon$ as the stop condition, when the condition is satisfied the loop ends. If not, the outer loop operation is performed again (step 2).

From the solving process of the above VMD algorithm, it can be concluded that the VMD algorithm adaptively decomposes the characteristic frequency of the original signal to get its frequency bandwidth. Through the termination condition to control the IMF and the center frequency to calculate repeatedly in the time-frequency domain of the signal. The adaptive decomposition process ends when the stop condition is satisfied.

#### 2.1.2. Sample Entropy

Sample entropy, which can better measure the complexity of time series, is widely used in signal analysis and processing.

Suppose that there are *N* pieces of data, and the time series of data sampling is defined as X=[x(n),n=1,2,…,N]. The theoretical derivation of the definition of sample entropy is as follows:

(1)According to the sampling time of the signal, a vector sequence based on time series is constructed, and the dimension of the vector sequence is *m*, $X_{m}\left(1\right),\ldots ,X_{m}\left(N-m+1\right)$. Each element in the vector sequence can be represented by the following array: $X_{m}\left(i\right)=\left\{x\left(i\right),x\left(i+1\right),\ldots ,x\left(i+m-1\right)\right\}$, 1≤i≤N−m+1. The array represents the continuous x values of the time series from i to m+i;(2)Define the distance between $X_{m}\left(i\right)$ and $X_{m}\left(j\right)$: $d\left[X_{m}\left(i\right),X_{m}\left(j\right)\right]$ is the absolute value of the difference between $X_{m}\left(i\right)$ and $X_{m}\left(j\right)$.
(10)$$d\left[X_{m}\left(i\right),X_{m}\left(j\right)\right]=\max_{k=0,\ldots ,m-1}\left(|x\left(i+k\right)-x\left(j+k\right)|\right)$$(3)For the constructed $d\left[X_{m}\left(i\right),X_{m}\left(j\right)\right]$, the number of j (1≤j≤N−m,j≠i) is calculated and marked as $B_{i}$, 1≤i≤N−m, $B_{i}$ is defined as follows:(11)$$B_{i}^{m}\left(r\right)=\frac{1}{N-m-1}B_{i}$$(4)Average $B^{\left(m\right)}\left(r\right)$ as Formula (12):(12)$$B^{\left(m\right)}\left(r\right)=\frac{1}{N-m}\sum_{i=1}^{N-m}B_{i}^{m}\left(r\right)$$(5)Update the vector dimension to m+1, and recalculate the number of distances and $d\left[X_{m}\left(i\right),X_{m}\left(j\right)\right]$≤r bands, where (1≤j≤N−m,j≠i) benchmark is marked as $A_{i}$. Define $A_{i}^{m}\left(r\right)$ and $A^{m}\left(r\right)$ as the following expressions:(13)$$A_{i}^{m}\left(r\right)=\frac{1}{N-m-1}A_{i}$$
(14)$$A^{m}\left(r\right)=\frac{1}{N-m}\sum_{i=1}^{N-m}A_{i}^{m}\left(r\right)$$

From the above steps, $B^{\left(m\right)}\left(r\right)$ is the probability of two sequences matching *m* points under the similar tolerance *r*, while $A^{m}\left(r\right)$ is the probability of two sequences matching m+1 points. Therefore, the definition of sample entropy is:(15)$$SampEn\left(m,r\right)=\underset{N\to \infty }{\lim}\left\{-\ln\left[\frac{A^{m}\left(r\right)}{B^{m}\left(r\right)}\right]\right\}$$

When *N* is a finite value, the following Formula can be used:(16)$$SampEn\left(m,r,N\right)=-\ln\left[\frac{A^{m}\left(r\right)}{B^{m}\left(r\right)}\right]$$

As can be seen from the above description, the sample entropy has the following characteristics:

(1)This feature quantity can avoid the disadvantage of approximate entropy, prevent the data length from being compared by itself and can make the operation results more accurate and consistent.(2)Comparing the two sequences, no matter what the scale of the two sequences is, if the *m* and *r* values are changed, the calculation results will not change.(3)In the process of signal acquisition, it is inevitable to lose some frames. For the sample entropy algorithm, the loss of a small part of data has no great impact on the overall structure. Sample entropy can restore the operation results of real data to the maximum.

In any algorithm involving parameter selection, the influence of parameters cannot be ignored. When calculating the sample entropy of the signal, the value of the parameters has the same important influence on the result of the sample entropy operation. According to the theoretical derivation in the previous section, the main parameters of sample entropy include embedding dimension *m*, similarity tolerance *r* and data points *N*., the indexes of these parameters are as follows:

(1)The embedded dimension *m* represents the dimension of the window function in the sample entropy algorithm, which is similar to the size of the window function in the Fourier transform. In most cases, m=1.2. When m>2, the deviation of the parameter value will result in the following: first, a large number of original data sets will be needed to increase the computational complexity of the algorithm; second, a too large *m* will affect the value of *r*, and there is *a* positive correlation between the two. When *m* is larger, *r* is larger, *r* will remove too much useful information.(2)Similarity capacity *r* is usually obtained based on (0.15~0.25)δ(x), where δ(x) represents the standard deviation of sampling. The *r* value is too large, resulting in invalid data redundancy; the *r* value is too small, resulting in a reduction in the amount of data in similar patterns.(3)*N* indicates the number of sampled data points, which is usually obtained from 100 to 6000.

### 2.2. Fault Identification Method Based on IBA-ISMO

The penalty factor ζ in ISMO and the parameter σ of Gaussian kernel function have a considerable influence on the classification result and running time. In this paper, the neural network algorithm based on the bat algorithm is selected to optimize the parameters. The algorithm has a good local search ability. By optimizing the bat algorithm, the shortcomings of the algorithm in the process of global optimization are improved and the global optimal solution of the parameters is obtained.

#### 2.2.1. The Improved Bat Algorithm

The bat algorithm is used to solve the optimal solution by simulating the feeding habits of bats through echolocation. The main idea of the algorithm is that each bat represents a solution in the feasible region, imitating the method of identifying the direction of bat sound waves. Bat individuals constantly emit pulses of a fixed range of frequencies and capture the sound waves reflected after the pulse collides with the target. The distance and position of the target are obtained according to the difference in pulse frequency and the time difference of senses to feel the pulse.

Let the dimension of search space be d-dimensional, and the relevant parameters emitted by bat *i* in the process of finding the optimal solution are pulse frequency $f_{i}$, velocity $v_{i}$, position $x_{i}$, transmitted pulse frequency $\left[f_{\min},f_{\max}\right]$ and the maximum number of iterations maxT. Therefore, the update Formula for the position of the bat at *t* moment is as follows:(17)$$f_{i}=f_{\min}+\beta (f_{\max}-f_{\min})$$
(18)$$v_{i}^{t}=v_{i}^{t-1}+(x_{i}^{t}-x_{*})f_{i}$$
(19)$$x_{i}^{t}=x_{i}^{t-1}+v_{i}^{t}$$
where β is a random number in [0,1], and $x_{*}$ is the optimal position of the current population.

In the process of searching for prey, each bat will adjust the loudness and pulse frequency of its sound wave according to the location of the target to improve the capture probability. In the process of getting closer to the target, the search area of the bat will gradually decrease. Therefore, when the loudness decreases below a certain fixed value, the frequency is rapidly increased to facilitate the faster acquisition of prey, and the changes of loudness and pulse in the process of catching prey can be obtained, as shown in the following Formula:(20)$$A_{i}^{t+1}=\beta A_{i}^{t}$$
(21)$$r_{i}^{t+1}=r_{i}^{0}[1-\exp(-\gamma t)]$$
where A represents the pulse loudness, γ>0 pulse represents the pulse frequency enhancement coefficient, and $r_{i}^{0}$ represents the initial pulse frequency.

As can be seen from the above Formula, when there is t→∞, there is $A_{i}^{t}\to 0$. When $A_{i}^{t}\to 0$, it means that the bat has found its prey at this time, and the iteration ends and no longer sends out pulses.

Bat algorithm has obvious advantages over other parameter optimization algorithms in global search ability and convergence speed, but it also has the disadvantage that individuals of the population are easy to fall into the local optimal solution. In order to solve this problem, this paper proposes an improved bat optimization algorithm by introducing a new variable w; namely, the adaptive weight factor, to measure the difference between the current position and the global optimal solution. In order to avoid the final solution vector falling into the local optimal situation to the greatest extent.

The calculation Formula of adaptive weight factor w is as follows:(22)$$w_{i}=\frac{\left(x_{i}-x_{*}\right)}{t+1}$$

By updating Formula (22) to:(23)$$v_{i}^{t}=v_{i}^{t-1}w_{i}+(x_{i}^{t}-x_{*})f_{i}$$

To sum up, the flow of the IBA algorithm is shown in Figure 2.

According to the characteristics of IBA algorithm, it is found that the parameters will affect the convergence speed of the algorithm itself and the accuracy of the optimal solution. For example, the parameters of this kind of group optimization algorithm need to be selected through strong experiment and experience, and either too large or too small parameters will affect the results, so the selected parameters are as follows:

Pulse enhancement coefficient γ=0.9, pulse frequency $f_{\max}=2$, $f_{\min}=0$; loudness coefficient $A_{0}=1.5$, initial pulse intensity $r_{0}=0.5$, algorithm population size n=50, dimension d=5, maximum iterations M=1000, adaptive weight factor $w_{\max}=0.9$,$w_{\min}=0.2$.

Because the IBA algorithm avoids the disadvantage of falling into the local optimal solution compared with the traditional BA algorithm, in order to verify whether the parameter selection of the IBA algorithm is reasonable, this section selects the Rastrigin function to test the global optimization performance of the IBA algorithm and selects the Ackley function to test the global convergence ability of the IBA algorithm.

(1)Rastrigin function


(24)
$$f\left(x\right)=\sum_{i=1}^{n}\left[x_{i}^{2}-10\cos\left(2\pi x_{i}\right)+10\right]$$


Among them, x∈[−5.12,5.12], i=1,2, and the overall shape of the function is similar to that of the hills, which proves that the algorithm has a good ability for global optimization. The function image is shown in Figure 3:

(2)Ackley function


(25)
$$f\left(x\right)=20+e-20\exp\left(-0.2\sqrt{\frac{1}{n}\sum_{i=1}^{n}x_{i}^{2}}\right)-\exp\left(\frac{1}{n}\sum_{i=1}^{n}\cos\left(2\pi x_{i}\right)\right)$$


Among them, x∈[−32.768,32.768], i=1,2. The closer f(x) is to 0, the stronger the global convergence ability of the algorithm is. The image of the IBA algorithm after applying this function is shown in Figure 4.

In order to digitize the image and show the global optimization ability and global convergence ability of the improved bat algorithm more intuitively, the IBA algorithm after parameter selection is run 15 times independently, and the test results shown in Table 1 are obtained.

As can be seen from the table, the standard deviation and average of the Rastrigin function and the Ackley function are both close to 0. Therefore, it has been proven that the IBA algorithm overcomes the disadvantages of the traditional BA algorithm and has a significant improvement in global convergence and global optimization.

#### 2.2.2. Improved Sequence Minimization Algorithm (ISMO)

As an algorithm in the SVM model, SMO algorithm essentially uses a very important functional relationship-kernel function. In this study, the Gaussian kernel function is improved to improve the efficiency of the SMO algorithm.

The accuracy of sample classification predicted by SMO should meet the following expression:(26)$$E_{N}[P\left(error\right)]\leq \frac{E_{N}{}_{\left[SV\right]}}{N}$$
where N represents the total number of training set samples and $E_{N}$ represents the expected value calculated through the training set samples. It can be seen from the Formula that when the number of samples in the training set is *N*, we can choose to reduce the number of support vectors to reduce the probability of operational errors and improve the application range of support vector machines. The control of the number of support vectors depends on the mapping relationship of the algorithm and the selection of algorithm parameters.

Based on the type of kernel function determined in the previous section, the Gaussian kernel function and the coefficient (1+m)(m>0) are as follows:(27)$$K\left(x,x_{i}\right)=\left(1+m\right)*\exp\left(-\gamma *{\left\|x-x_{i}\right\|}^{2}\right)$$

According to Formula (27), the Gaussian kernel coefficient is magnified by (1+m) times, and the number of support vectors and the number of samples on the boundary are reduced by increasing the absolute value of the quadratic coefficient in $Q^{v}$, which can effectively reduce the classification error rate. By reducing the solution vectors in data samples that meet the KKT boundary conditions, the time complexity of the algorithm is reduced, and the SMO classification accuracy and application range are improved. The improved SMO algorithm is named ISMO.

Based on the application background of the system in engineering practice, the acquisition and analysis system take the vibration signal as the original signal, and the original signal has the characteristics of small sample and non-linearity. The ISMO model is selected to classify the vibration signal eigenvector obtained in the previous chapter, and the mapping of eigenvector from linear inseparable to linear separable is completed, which enhances the classification accuracy and application range of the algorithm.

## 3. Experiment and Analysis

### 3.1. Extracted Features of Rolling Bearing Signals

A variational mode decomposition algorithm is an adaptive signal decomposition algorithm. By using this method, not only can part of the noise signal be removed, but also the information of the signal will not be lost, and the characteristic components of the original signal can be preserved as much as possible, so the VMD algorithm is chosen to preprocess the original signal. Sample entropy is a kind of eigenvalue used to measure the complexity of time series, which is improved on the basis of other entropy values, so it also has the characteristics of anti-noise, so sample entropy is chosen as the eigenvalue of the IMF signal. Therefore, this paper proposes a method of feature extraction by combining the VMD algorithm with sample entropy. The detailed flow chart is shown in Figure 5.

In order to verify the effectiveness of the feature extraction algorithm based on VMD and sample entropy proposed in this study, this section uses the bearing data in the CRWU database for related experiments [20]. The data set is mainly composed of the following data: drive acceleration data, fan segment acceleration data, basic acceleration data and speed data. The experimental system consists of test bearings, torque sensors, control motors with different functions and programmable controllers. The test bench is shown in Figure 6.

In the experiments, the sampling frequency is 12 kHz, the motor speed is 1797 r/min, and the fault state bearing damage diameter is 0.1778 mm. The bearing states selected in this experiment include normal state, inner ring fault, outer ring fault and roller fault, and the number of sampling points of each sample is 6000. The original sampling signals of the four states of the bearing are shown in Figure 7:

From the original vibration signal shown in Figure 7, it can be seen that there are great differences in the vibration period and amplitude of the bearing in different states. The vibration signals of the three fault states all confirm the above analysis of the vibration signal that there is a periodic abnormal signal, and there is little difference in amplitude in different periods in the same state.

The vibration signals of four states in Figure 7 are decomposed by variational mode decomposition. Because the variational mode algorithm has the advantage of removing some redundant component information, as shown in Figure 8, four IMF component informations are obtained according to different time–frequency domain characteristics.

The sample entropy of the decomposed components is calculated, and four groups of data are randomly selected from each state, as shown in Table 2. As can be seen from Table 2, there are obvious differences in the sample entropy of each modal component after the VMD decomposition of vibration signals in different states. Therefore, the sample entropy index based on VMD decomposition can be used as the eigenvalue of the bearing. The total number of samples obtained according to the above process is 350 × 4 = 1400.

From the characteristic components of sample entropy in Table 2, we can also see that the sample entropy eigenvalues of the four intrinsic mode functions in different states are quite different. For example, in the normal state, the eigenvalue of IMF1 is the lowest and IMF4 is the highest among the four states; the IMF1 component has the highest eigenvalue in the inner ring fault state; the IMF1 and IMF2 have relatively high eigenvalues in the rolling body fault state; in the outer ring fault state, the eigenvalue is the lowest among the four states and the highest in the four states of IMF2. From the above analysis, it can be concluded that the vibration signal is decomposed into the IMF component by the VMD algorithm, and the sample entropy characteristic value of the IMF component has a high degree of identification and discrimination. Therefore, the sample entropy characteristic index based on VMD decomposition can be used as the eigenvalue of the bearing.

### 3.2. Result of Fault Identification Based on IBA-SMO Algorithm

In the experiment, the feature extracted sample set is divided into a training set and verification set, and the training set is input to the ISMO model for training. According to the improved bat algorithm (IBA), the optimal penalty factor and “Gaussian kernel function parameter” of the ISMO model are obtained while training the ISMO model parameters of the training set samples. The verification set validates the trained model and verifies its ability to classify fault types. The set parameters of IBA algorithm are shown in Table 3.

Three hundred sets of samples are selected from each group as the training, and the rest as the prediction set. Then, all the training sets are input into the IBA-ISMO algorithm, and the values of the penalty factor ζ and kernel function parameter σ of the best fitness are obtained by IBA algorithm. The iterative process and the changing process of the evaluation function are shown in Figure 9. As can be seen from Figure 9, the evaluation function in IBA algorithm constantly calculates the fitness value produced by the matching of different penalty factor γ and kernel function parameter σ, and the fitness increases with the increase of the number of iterations until the optimal fitness value is obtained when the maximum number of iterations is close to the maximum number of iterations. At this time, the output fitness σ=87.63, γ=5.78.

In order to better prove that the improved sequence minimum optimization algorithm has a significant improvement in classification accuracy, firstly, the penalty factor and kernel function parameter obtained by IBA algorithm are input into the ISMO model as input parameters, and the optimal classification surface of the sample set is obtained. As shown in Figure 10, the optimal classification plane has completely separated different faults.

Figure 11 shows that IBA optimizes the fault identification accuracy of the traditional SMO model. It can be seen that there are misjudgments in some test sets, although the overall fault identification rate is 95.5%. Using the IBA-ISMO algorithm introduced in this study to re-train the training set samples and re-input the test set samples into the model derived by the IBA-ISMO algorithm, and the verification results are shown in Figure 12. Among them, class labels from 1 to 4 represent the normal state, inner ring fault, rolling fault and outer ring fault, respectively.

The results of the validation set input into the different models are shown in Table 4. It can be seen from Table 4 that the accuracy of the IBA-ISMO model is significantly higher than that of other models except PSO-ISMO, so it shows that the IBA-ISMO model can better identify the faults and can be effectively applied to the fault diagnosis of rolling bearings.

## 4. Conclusions

Aiming at the fault characteristics of rolling bearings, a feature extraction algorithm based on variational modal decomposition and sample entropy has been proposed, and most importantly, an improved fault identification method, IBA-ISMO, was proposed in this study. Using the CWRU data set as a sample set to verify the IBA-ISMO, it is confirmed that the method has a higher fault recognition rate than the comparison method, while the effectiveness of feature extraction for instability vibration signals has been indirectly proven. The main work of this research is as follows:(1)The VMD algorithm is employed to adaptively decompose the characteristic frequency of the original signal to obtain its specific frequency bandwidth, and the sample entropy is used to extract the characteristics of the IMF component, highlighting the fault information.(2)An improved bat optimization is designed to optimize the classifier’s parameters, which avoids the disadvantages of falling into local optimal solutions compared with the traditional BA algorithm.(3)The research improves the Gaussian kernel function coefficient of the traditional SMO method, which effectively reduces the classification error rate and optimizes the algorithm’s time complexity by reducing the solution vectors that meet the boundary conditions in the data samples.

It should be noted that effective features are very beneficial for fault diagnosis. In this study, only the variational mode decomposition is performed on the signal, and the sample entropy of the component is used as the fault feature. The follow-up research will focus on the fault characteristics and fault phenomena. On this basis, an in-depth analysis of the interpretability of deep learning methods will be carried out.

## Figures and Tables

**Figure 1 sensors-23-00991-f001:**
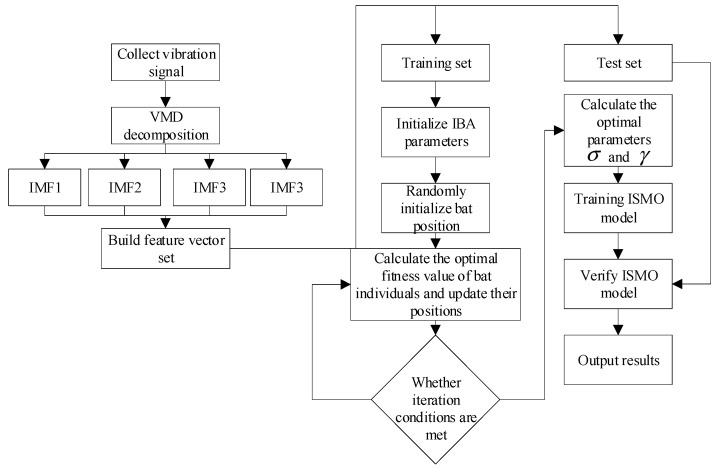
The model of fault diagnosis.

**Figure 2 sensors-23-00991-f002:**
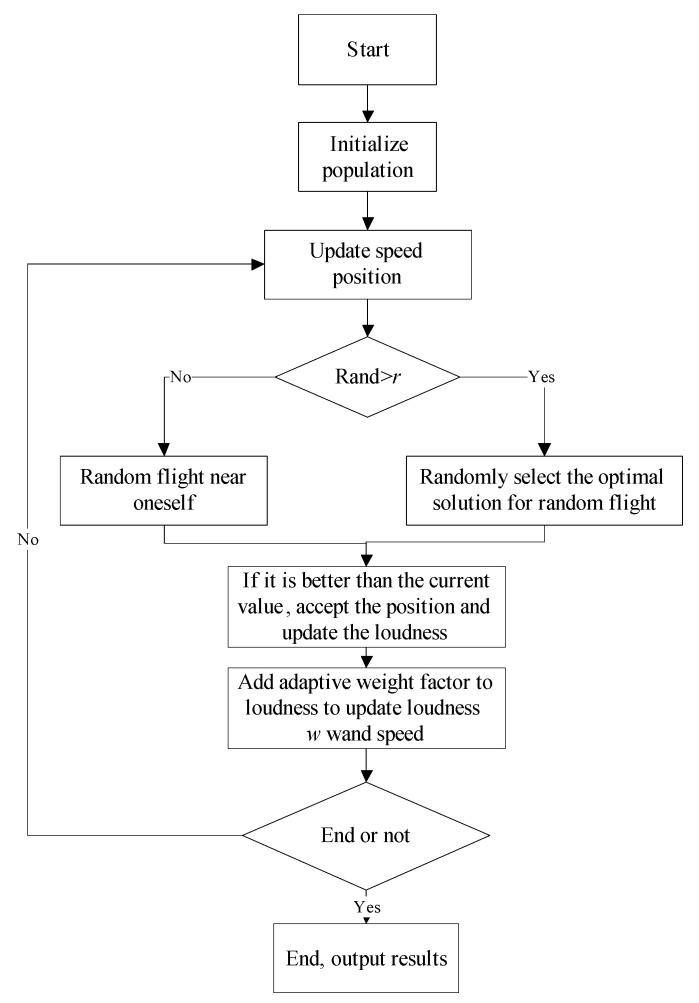
The flowchart of IBA algorithm.

**Figure 3 sensors-23-00991-f003:**
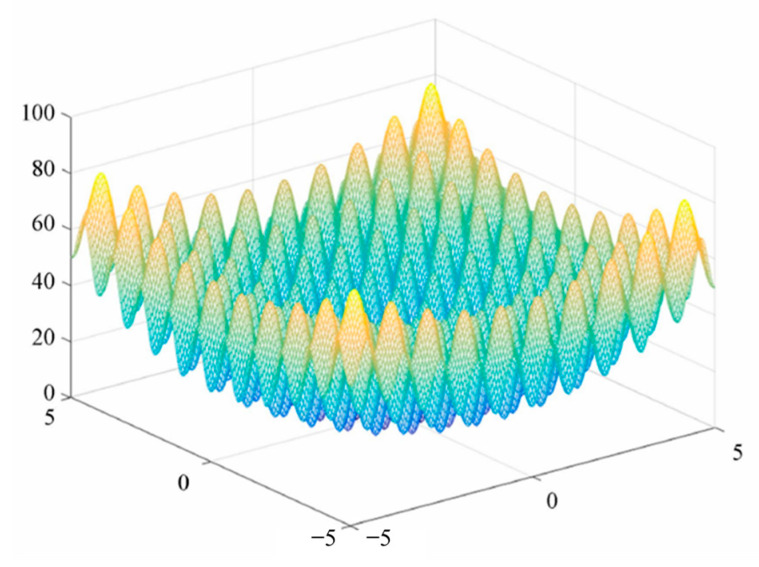
Rastrigin function.

**Figure 4 sensors-23-00991-f004:**
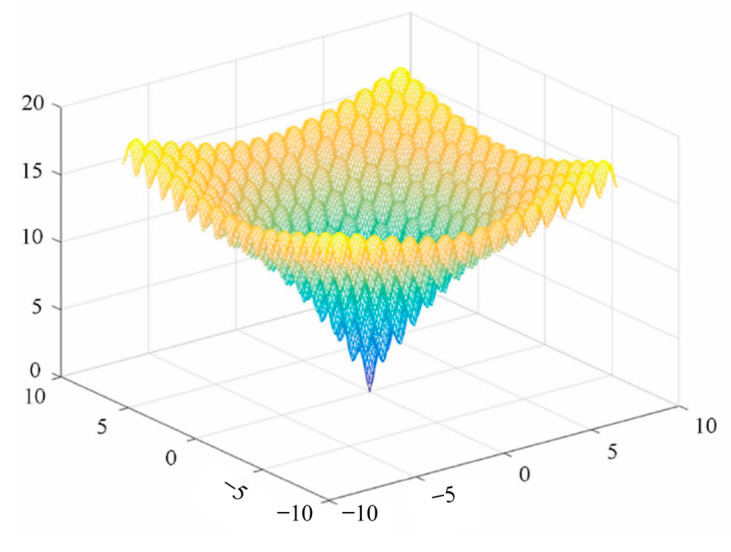
Ackley function image.

**Figure 5 sensors-23-00991-f005:**
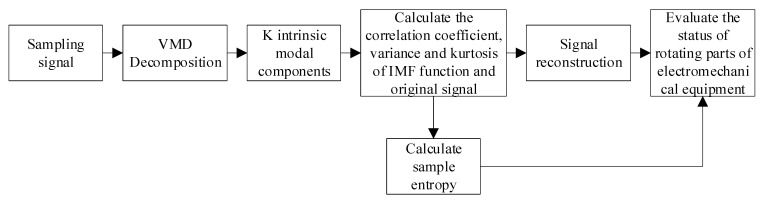
Flow chart of feature extraction.

**Figure 6 sensors-23-00991-f006:**
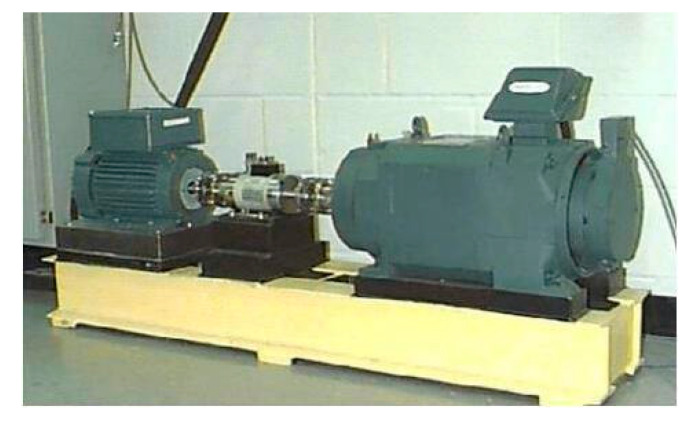
Bearing simulation failure test bench of Western Reserve University.

**Figure 7 sensors-23-00991-f007:**
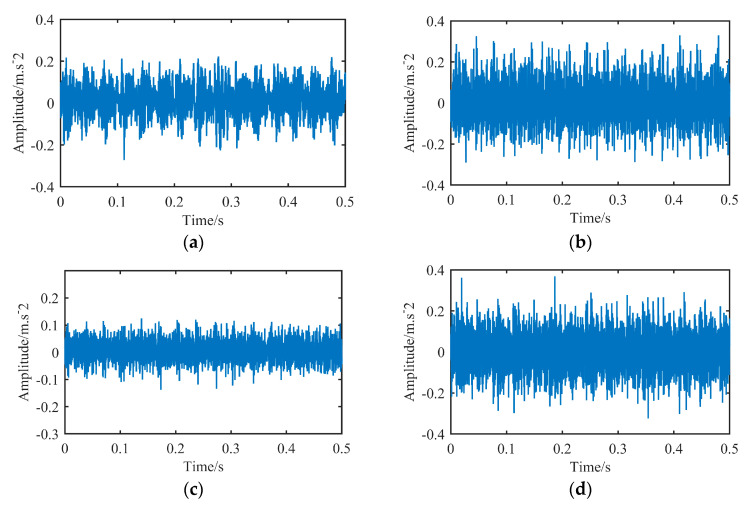
Vibration signals of rolling bearings in four states. (**a**) Normal state. (**b**) Inner ring fault. (**c**) Rolling body fault. (**d**) Outer ring fault.

**Figure 8 sensors-23-00991-f008:**
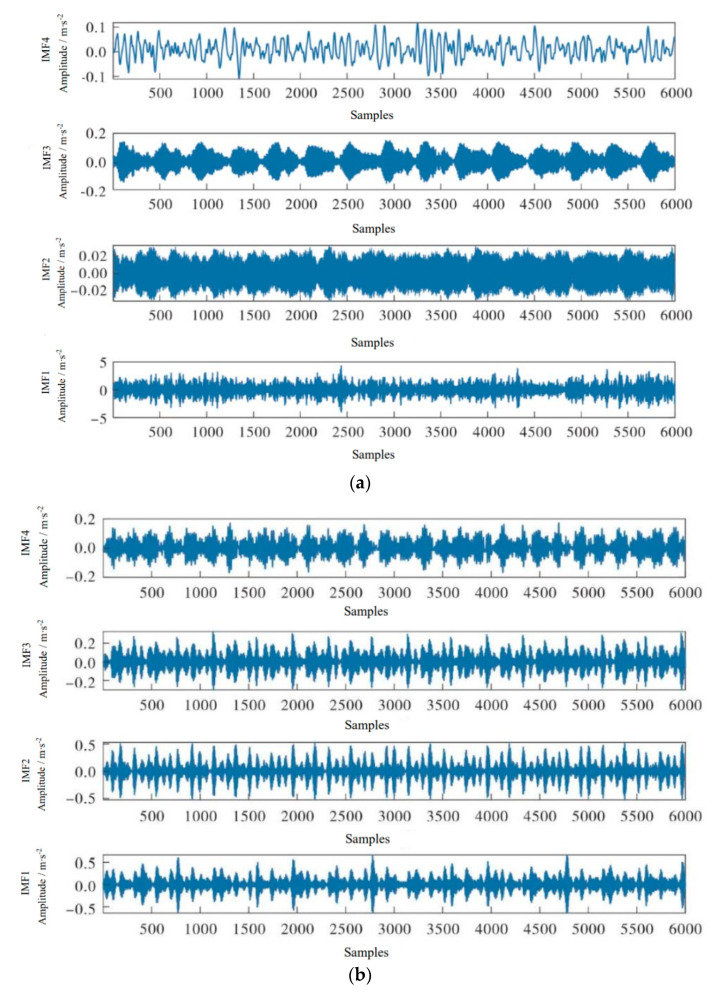
VMD decomposition results of four states. (**a**) Normal. (**b**) Inner ring fault. (**c**) Rolling body fault. (**d**) Outer ring fault.

**Figure 9 sensors-23-00991-f009:**
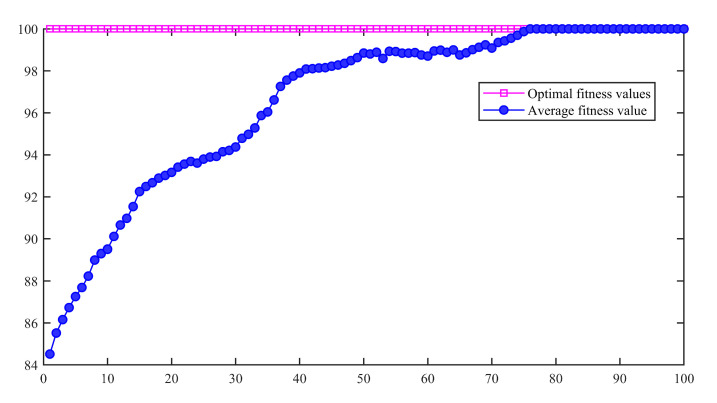
The optimal fitness curve of the improved bat algorithm.

**Figure 10 sensors-23-00991-f010:**
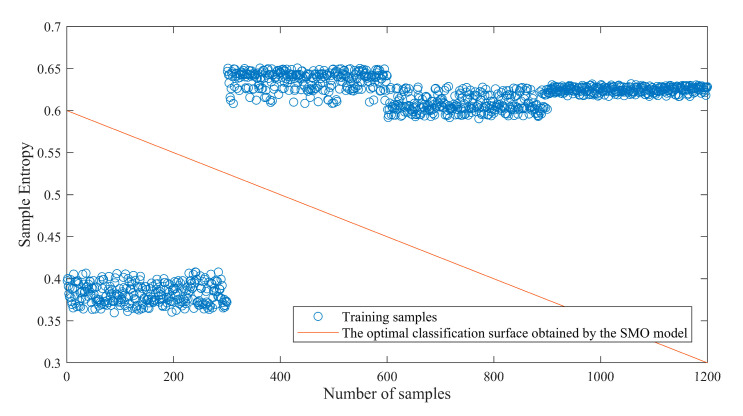
The optimal classification surface of the SMO model.

**Figure 11 sensors-23-00991-f011:**
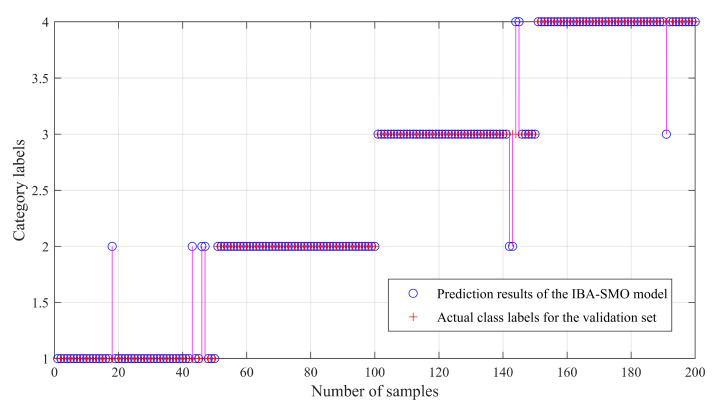
IBA-SMO Model verification.

**Figure 12 sensors-23-00991-f012:**
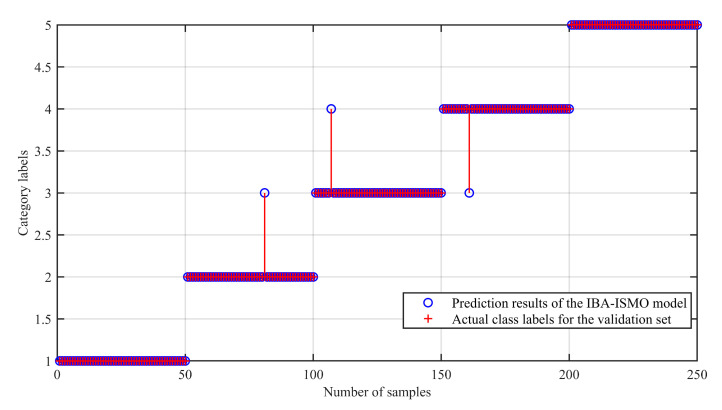
IBA-ISMO Model verification.

**Table 1 sensors-23-00991-t001:** Test function results.

Function	Algorithm	Optimal Value	Average Value	Standard Deviation
Rastrigin	IBA	0	7.11 × 10^−16^	1.50 × 10^−15^
Ackley	IBA	4.26 × 10^−14^	5.97 × 10^−13^	8.33 × 10^−13^

**Table 2 sensors-23-00991-t002:** Sample entropy of some samples.

Status	Sample Entropy Features			
	IMF1	IMF2	IMF3	IMF4
Normal	0.270606	0.538289	0.266344	0.756758
	0.278523	0.547741	0.261999	0.741383
	0.27207	0.556223	0.234584	0.815750
	0.271301	0.543783	0.229493	0.520792
Inner ring fault	0.583038	0.507441	0.245161	0.276670
	0.592259	0.510691	0.239027	0.287463
	0.585477	0.483586	0.304042	0.234883
	0.586028	0.487997	0.317364	0.219320
Rolling element fault	0.427306	0.609396	0.267466	0.155322
	0.398368	0.512955	0.240769	0.177347
	0.589774	0.482116	0.304627	0.163799
	0.582455	0.473968	0.281762	0.206347
Outer ring fault	0.427306	0.609396	0.267466	0.155322
	0.398368	0.595303	0.228638	0.157244
	0.578411	0.495655	0.192259	0.119652
	0.579525	0.506227	0.204141	0.142848

**Table 3 sensors-23-00991-t003:** Parameter setting for IBA.

Population Size	Population Dimension	Number of Iterations	Loudness Factor	Search Range of σ	Search Range of γ
50	5	100	1.5	1~100	1~100

**Table 4 sensors-23-00991-t004:** Model recognition results.

Model Types	Inner Ring Fault	Rolling Fault	Outer Ring Fault	Normal State	Overall Accuracy Rate	Training Time (s)
BA-SMO	90%	92%	90%	96%	92%	5.94
GA-SMO	90%	96%	90%	90%	91.5%	6.65
PSO-SMO	96%	98%	96%	94%	96%	8.99
BA-ISMO	94%	100%	96%	98%	96%	3.35
GA-ISMO	92%	98%	94%	98%	95.5%	4.52
PSO-ISMO	100%	98%	96%	100%	98.5%	7.85
IBA-SMO	92%	100%	92%	98%	95.5%	6.36
IBA-ISMO	100%	98%	98%	98%	98.5%	5.58

## Data Availability

This data comes from the public data set of Case Western Reserve University, and the link is “https://engineering.case.edu/bearingdatacenter/apparatus-and-procedures (accessed on 7 August 2022)”.

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
