# Peer review of "The IBA-ISMO Method for Rolling Bearing Fault Diagnosis Based on VMD-Sample Entropy"

_sensors, 2023, doi:10.3390/s23020991_

Round 1

Reviewer 1 Report

This paper aiming at the fault characteristics of rolling bearings, a feature extraction algorithm based on variational modal decomposition and sample entropy has been proposed. Mean- while, an optimal identification method named IBA-ISMO also has been designed. It is interesting. However, it needs to be modified before publishing on Sensors. Some comments can be summarized as follows:

1.     Using the optimization algorithm to optimize the parameters of sample entropy, better parameters can be obtained.

2.     IBA needs to be compared with other optimization algorithms to illustrate the advantages of this algorithm

3.     What are the parameters of the IBA algorithm based on?

4.     It is necessary to compare the performance of IBA and BA algorithm in Rastrigin function and Ack-ley function, so as to show that IBA algorithm is superior to BA algorithm.

5.     Training time and test time, training accuracy and test accuracy should be put together in a clearer way

6.     The feature extracted in this paper should be compared with the traditional fault feature to illustrate the effectiveness of the feature extracted in this paper.

7.     Why not use deep learning or other methods for troubleshooting, and what are the advantages of the methods presented in this article? This needs to be illustrated by comparison with other methods.

Reviewer 2 Report

1. The abstract needs to revise and improve in a way that the key idea to the existing challenges be well explained.

2.The clarity of some figures in the text is too low, which makes it difficult to see the effects of different methods ,such as Figure 7, 9.

3.The workload of the experimental link in this paper is too little. It is suggested to compare and analyze the recognition accuracy with other existing methods to highlight the effectiveness of the proposed method.

4.The conclusion of this paper does not well summarize the beneficial results obtained in this paper.

Round 2

Reviewer 2 Report

none